# Pectin–Zeolite-Based Wound Dressings with Controlled Albumin Release

**DOI:** 10.3390/polym14030460

**Published:** 2022-01-24

**Authors:** Banu Kocaaga, Ozge Kurkcuoglu, Melkon Tatlier, Gizem Dinler-Doganay, Saime Batirel, Fatma Seniha Güner

**Affiliations:** 1Department of Chemical Engineering, Istanbul Technical University, Maslak, Istanbul 34469, Turkey; bkocaaga@itu.edu.tr (B.K.); olevitas@itu.edu.tr (O.K.); tatlierm@itu.edu.tr (M.T.); 2Department of Molecular Biology and Genetics, Istanbul Technical University, Maslak, Istanbul 34469, Turkey; gddoganay@itu.edu.tr; 3School of Medicine, Marmara University, Istanbul 34854, Turkey; saime.batirel@marmara.edu.tr

**Keywords:** pectin, wound dressing, zeolite, albumin

## Abstract

Hypoalbuminemia can lead to poor and delayed wound healing, while it is also associated with acute myocardial infarction, heart failure, malignancies, and COVID-19. In elective surgery, patients with low albumin have high risks of postoperative wound complications. Here, we propose a novel cost-effective wound dressing material based on low-methoxy pectin and NaA-zeolite particles with controlled albumin release properties. We focused on both albumin adsorption and release phenomena for wounds with excess exudate. Firstly, we investigated albumin dynamics and calculated electrostatic surfaces at experimental pH values in water by using molecular dynamics methods. Then, we studied in detail pectin–zeolite hydrogels with both adsorption and diffusion into membrane methods using different pH values and albumin concentrations. To understand if uploaded albumin molecules preserved their secondary conformation in different formulations, we monitored the effect of pH and albumin concentration on the conformational changes in albumin after it was released from the hydrogels by using CD-UV spectroscopy analyses. Our results indicate that at pH 6.4, BSA-containing films preserved the protein’s folded structure while the protein was being released to the external buffer solutions. In vitro wound healing assay indicated that albumin-loaded hydrogels showed no toxic effects on the fibroblast cells.

## 1. Introduction

Hypoalbuminemia is a disorder caused by low albumin production rates or an increased loss due to several possible reasons [1]. It is highly associated with both acute and chronic wound-related complications [2]. During the process of wound healing, a large amount of plasma proteins, including albumin, globulin, and fibrinogen, is provided by the exudate to support the cellular activities necessary for repairing and remodeling of the tissue [3,4]. In addition to fatty acids, hormones, drugs, and enzymes, albumin also carries zinc, which is required for tissue growth, immune functions, collagen synthesis, and clear healing progress [5]. There has been a recent interest in the use of zinc oxide nanoparticles against bacterial infection and for acceleration of wound healing [6,7,8].

Exudate is generally produced during the inflammatory and proliferative stages of the healing process. In acute wounds, exudate production reduces over time; on the other hand, in chronic and burn wounds and wound dehiscence, large amounts of exudate can be produced, which can lead to a significant amount of albumin loss [9]. Bacterial contamination has a special importance to the chronic inflammation of wounds that do not heal [6,10]. Reduction in serum albumin levels can lead to prolonged inflammation, poor and delayed wound healing, tissue oedema, elevated levels of reactive oxygen species, muscle atrophy, and an increase in morbidity and even mortality, especially in major burn patients [11,12,13,14,15].

Albumin infusion and normal protein diet administration methods are commonly practiced in the wound treatments of hypoalbuminemia patients [16]. On the other hand, some clinical evidence suggests that albumin administration to critically ill patients with hypovolemia from injury or surgery and burns should be restricted to the indications for cases where albumin treatment is effective [16]. Here, the systemic exposure to albumin during the treatment is implied to increase the mortality rates of these patients. To help the healing process while reducing albumin-related mortality and side effects, albumin can be delivered directly and locally to the wounded area.

Hydrogel systems have been broadly used in the development of delivery systems for drugs [17,18] and protein [19]. Because of their hydrophilic structure [10], natural polysaccharides in hydrogel form such as chitosan [6,20,21], alginate [10,22], starch [23], hyaluronic acid, and pectin [24] have been widely used in the fabrication of dressings for burned, exuding, and bleeding wounds. Pectin, one of the most abundant polysaccharides, is a promising biomaterial for tissue engineering and drug delivery systems due to its good biocompatibility, biodegradability, and antimicrobial activity [25]. Due to their anionic character, pH-sensitive pectin hydrogel films appear as good candidates for controlled release systems to treat wounds, where pH changes according to the healing progress. We recently showed the controlled drug delivery ability of pectin-based hydrogels [26,27,28,29], where the mechanical and thermal properties can be adjusted with the addition of other polymeric or inorganic materials [25,29,30]. These hydrogels combined a variety of properties desired for a wound dressing, where high oxygen permeability, moisture-holding capacity, and antimicrobial properties are a few from the long list.

In this study, we report the synthesis of pectin-based hydrogel films with the ability to deliver albumin in a controlled manner without altering the folded protein structure. To improve the oxygen permeability as well as stall the decomposition of the pectin hydrogel, zeolite is used, as in our previous study [25]. Bovine serum albumin (BSA) is chosen as the model, since it has a structural similarity to human serum albumin [31]. We investigate in detail the effect of BSA loading pH and concentration on BSA release behavior and protein native structure. To the best of our knowledge, this is the first study to report controlled albumin delivery while preserving its biological activity from a hydrogel matrix, with excellent wound dressing properties for the treatment of wounds with excess exudate.

## 2. Materials and Methods

Amidated low-methoxyl pectin (LM, degree of esterification = 49%) is provided from Herbstreith & Fox Company (Neuenbürg, Germany). Bovine serum albumin (BSA), Glycerol (purity-99%), CaCl_2_.2H_2_O, and TRIS (2-Amino-2-(hydroxymethyl)propane-1,3-diol) are supplied from Sigma Aldrich (St. Louis, MI, USA). Sodium silicate (Merck Darmsdadt, Germany), granular sodium aluminate (Riedel-de Haen, Hannover, Germany), and NaOH pellets (Carlo Erba, Val-de-Reuil, France) are purchased for zeolite synthesis. All chemicals are of analytical grade and are not further purified. Ultra-deionized water is used throughout the experiments.

### 2.1. Zeolite-A Synthesis

Zeolite-A is synthesized in Na form (NaA) from a gel-type reaction mixture with a molar composition of 3.165 Na_2_O:1 Al_2_O_3_:1.926 SiO_2_:128 H_2_O [32]. Sodium silicate, granular sodium aluminate, NaOH pellets, and deionized water are utilized as reagents. The reaction mixture is loaded into a polypropylene vessel, and the synthesis is carried out in an oven at 99 °C for 3.5 h.

Some of the zeolite-NaA (alternatively named Na-zeolite) obtained is ion-exchanged to prepare zeolite-A in Zn form (alternatively named Zn-zeolite). First, 1 L of 2 M ZnCl_2_ solution is prepared, followed by adding 80 g of Na-zeolite, and the mixture is stirred at 200 rpm in darkness for 3 days at room temperature, as detailed in [33]. Solid material is filtered from the mixture, and it is analyzed using inductively coupled plasma (ICP) to control the exchange of Na in zeolite-A to Zn ions.

### 2.2. Preparation of BSA-Loaded Hydrogels

Pectin–zeolite hydrogels are prepared using the ionotropic gelation method, where CaCl_2_ is used as a cross-linker and glycerol as a plasticizer [25]. In total, 30 mg of Na-zeolite is used in the experiments. BSA is loaded to pectin–zeolite hydrogels at different concentrations (10, 25, 50, 75, or 100 mg BSA/g film) and pH values (4.3, 4.7, or 6.4). Two loading methods, mixing and adsorption, are investigated in this study. BSA-loaded pectin–zeolite hydrogels are coded according to their compositions in the Tα- β notation and are listed in Table 1. Here, T is the type of method (A, adsorption; M, mixing), α denotes the pH of the BSA dissolution media, and β the concentration of BSA in mg/g-film. PZ-30 and PZ-50 represent pectin hydrogel with 30 and 50 mg zeolites without BSA, respectively.

In the mixing method, hydrogel films are prepared based on diffusion into membrane-controlled system. BSA-loaded Na-zeolite is used for core particles, which are then surrounded by cross-linked LM-pectin chains as a membrane that is permeable to both BSA and water. Thus, zeolite-Na serves as a saturated reservoir for BSA. Before experiments, the zeolite particles are dehydrated at 120 °C under vacuum overnight. Zeolite-BSA suspensions are stirred on an orbital shaker at 25 ± 1 °C and 100 rpm for 24 h in a dark medium. BSA loading patterns to zeolite particles are determined using LAMBDA 1050 UV/Vis Spectrophotometer in TRIS buffer solution at 279 nm. Then, 6 mL glycerol solution (5% *w*/*w*) is added slowly into LM-pectin solution in water (2% *w*/*w*) and stirred on a magnetic stirrer at 150 rpm for 2 h. BSA-loaded zeolite suspension is added to the pectin–glycerol solution. The interaction between polyelectrolytes and proteins requires time to achieve a high conversion of the complex [34]. Pectin–glycerol and BSA-loaded zeolite suspension are stirred at 150 rpm and 27 ± 1 °C for 24 h in a dark medium. Then, the suspension is poured onto 10 mL CaCl_2_ solution in a petri plate where gelled films are formed spontaneously. The mixture is dried at 27 ± 1 °C on an orbital shaker to obtain a dry BSA-loaded pectin–zeolite hydrogel film [35].

In the adsorption method, zeolite-Na particles in ultra-deionized water are mixed in an ultra-sonic bath for 15 min and are added to pectin–glycerol solution immediately at the end of 2 h. The resulting suspension is mixed overnight in a dark medium. Hydrogel films are prepared without BSA, as described in the mixing method. After the films are dried in petri plates where complete evaporation of water is achieved, BSA solution is added to dry hydrogel films. Hydrogel films with BSA solution are left on an orbital shaker for 24 h to complete adsorption followed by the drying process.

### 2.3. Characterization of Hydrogel Films and BSA Adsorbtion and Release Behavior

#### 2.3.1. Zeta Potential Measurement of Suspensions

The zeta potential is determined for pectin (2 wt%), pectin–glycerol, and pectin–glycerol–zeolite suspensions at various pH values (6.4, 4.7, or 4.3). All solutions are prepared for measurement at identical conditions for preparing the hydrogel film. The measurements are performed using laser Doppler micro-electrophoresis with a Zetasizer Nano Series (Malvern Instruments, Worcestershire, UK) in conjunction with capillary cell. The zeta potential is calculated using mean electrophoretic mobility via Smoluchowski’s equation.

#### 2.3.2. Rheological Analysis

Rheological properties of solutions and hydrogels are determined in Anton Paar Physica MCR 301 (Anton-Paar, Graz, Austria) with CP25 and PP25 measuring plates at 25 °C. The sample temperature is controlled by a Viscotherm VT2. Experiments are performed by putting cross-linked wet hydrogel samples into the rheometer after 6 hr of mixing in the CaCl_2_ solution. GAP is 2.7 ± 0.2 mm. First, a strain sweep at an angular frequency of 10 rad/s is performed for each sample to specify the linear viscoelastic region. Then, a frequency sweep test is performed over a range of 0.1 to 100 rad/s to determine the storage modulus (G′) and loss modulus (G′′).

#### 2.3.3. FT-IR Analysis

The FT-IR spectra of the hydrogel films are obtained in the range of 4000–400 cm^−1^ using KBr pellet or attenuated total reflectance (ATR) technique with the help of PerkinElmer Spectrum One FT-IR Spectrometer (Perkin–Elmer Inc., Beaconsfield, United Kingdom) at room temperature. All measurements are triplicated.

#### 2.3.4. Morphological Analysis

The surface and cross-section morphologies of dried films are investigated using scanning electron microscopy (SEM) (SEM, JSM-6480LV; Jeol, Tokyo, Japan). Hydrogel samples are immersed in nitrogen for 5 min. The frozen samples are cross-fractured manually using cold tweezers.

To investigate the surface roughness of dry hydrogel films, Nanosurf atomic force microscopy (AFM) (NaioAFM, Basel, Switzerland) is used in tapping mode for 50 × 50 mm^2^ surface area. The measurements are performed under air with dried samples. Three independent pieces of 12.5 × 12.5 mm^2^ on each sample are investigated for the estimation of the surface roughness.

#### 2.3.5. Contact Angle Measurements

Static water contact angle measurements are made using KSV CAM200 goniometer (KSV Instrument Ltd., Helsinki, Finland) at room temperature by placing 5 μL of distilled water on the hydrogel surface using sessile drop method.

#### 2.3.6. Swelling Degree

To determine the swelling degree, circular pieces with diameter of 12 mm are cut from the dry hydrogel films. They are immersed in 25 mL pH 6.4 TRIS buffer at room temperature in a petri plate. After specific periods, samples are removed from the buffer, weighed, and then immediately returned into the buffer solution. Before each weighing, excess moisture on the sample surface is removed by gently placing the film between two sheets of filter paper. The equilibrium rate of swelling for each composite is calculated by:(1)$$\mathrm{Swelling}\;\mathrm{degree}=\frac{W_{s}-W_{d}}{W_{d}}$$
where *W_s_* is the weight of the product after hydration and *W_d_* is the weight of dried samples.

#### 2.3.7. In Vitro Drug Release from Hydrogels

In vitro drug release patterns of the films are evaluated in TRIS buffer at a pH of 6.4. BSA-loaded dry hydrogel film samples are put in vials containing buffer solution and are shaken on an orbital shaker at 100 rpm at 25 ± 1 °C. The concentration of BSA release is monitored with UV/Vis Spectrophotometer (Perkin Elmer Corp, Waltham, MA, USA) at 279 nm. Solution samples are withdrawn from the quartz cuvette vial at regular intervals and are returned to the vial immediately after their absorbance is measured. The drug concentration is determined with Lambert–Beer’s law. The experiments are performed in triplicate under the same conditions.

#### 2.3.8. Statistical Analysis

Statistical analysis of data is performed using Minitab16 (Minitab Inc., State College, Pennsylvania, PA, USA). Analysis for rheological and BSA adsorption data is carried out using two-way ANOVA where pH and BSA concentration are the independent variables. Hydrogel swelling data are analyzed using three-way ANOVA method considering pH, BSA concentration, and hydrogel synthesis method. BSA release data from hydrogel samples are assessed with four-way ANOVA considering the effect of pH, BSA concentration, hydrogel synthesis method, and zeolite addition. All analyses with ANOVA method are followed by Tukey’s test for pairwise and simultaneous comparison of the independent variables with a 95% confidence level.

#### 2.3.9. Circular Dichroism (CD) Measurements

Circular dichroism (CD) (Jasco 810 spectropolarimeter (Jasco, Tokyo, Japan) is used to reveal any changes in the secondary structure of BSA molecules [36].

Selected BSA-loaded hydrogel films are put in vials containing TRIS buffer solution (pH 6.4) and are shaken on an orbital shaker at 100 rpm, 25 ± 1 °C overnight. CD measurements of the solutions in vials are carried out under continuous nitrogen atmosphere at room temperature for far-UV region (190–260 nm), using a quartz cell with a path length of 0.1 cm. Continuous-scan mode is used for all samples, where three scans are accumulated for analysis. A 50 nm/min scan rate is used throughout the experiments. Reported UV-CD spectra are baseline subtracted for TRIS buffer (pH 6.4), and results are taken as ellipticity in mdeg.

#### 2.3.10. In Vitro Wound Healing Assay

We perform a wound healing assay using fibroblast cells to assess the wound healing properties of the hydrogels. PCS-201-012 human dermal fibroblast cells (American Tissue Culture Collection, Manassas, VA, USA) are seeded into collagen-coated 24-well tissue culture plates and are then maintained at 37 °C and 5% CO_2_. Following the attachment of cells, a scratch is made within the monolayer of the cells with a pipette tip. The cells are treated with different types of hydrogels or kept only in the culture medium (as a control group) for 48 h. Photomicrographs of scratched area are taken by a microscope (Carl Zeiss, Jena, Germany) at 0 h, 24 h, and 48 h of incubation period without changing the cell culture medium. The assays are performed at least in duplicate.

#### 2.3.11. Molecular Dynamics Simulations

We investigate structural dynamics and electrostatic surface properties of human serum albumin (PDB ID 4LB9) dynamics in water. Human and bovine serum albumins have high sequence and structure similarities [31]; therefore, results obtained from MD simulations can be used in interpreting the loading/release experiments conducted in this study. Full-atom molecular dynamics (MD) simulations of human serum albumin are performed in explicit water using the CHARMM36 force field [36] in the NAMD 2.13 package (University of Illinois at Urbana-Champaign, Urbana, IL, USA) [37]. TIP3P water model is used in all calculations. Simulation box with periodic boundary conditions is first subjected to energy minimization of 1000 steps of conjugate gradient to relax steric clashes. Then, simulations are carried out using NPT ensemble, with a temperature of 310 K and ionic strength of 150 mM. Electrostatic interactions are calculated using particle mesh Ewald method, with a cut-off distance of 12.0 Å. MD simulations are run for 55 ns with a time step of 2 fs using the SHAKE algorithm [38].

Electrostatic surface distributions of human serum albumin structures are determined using adaptive Poisson–Boltzmann solver approach in PROPKA server [39]. Electrostatic surface properties are calculated at experimental pH values of 4.3, 4.7, and 6.4. Results are visualized on three-dimensional crystal structures with PYMOL Molecular Graphics System (De Lano Scientific, San Carlos, CA, USA) [40].

## 3. Results and Discussion

In order to understand BSA behavior inside the pectin–zeolite matrix, we first calculate the electrostatic properties of the dynamic serum albumin and the zeta potential of all hydrogel prepared solutions. Then, we conduct rheological, topological, and morphological analysis for the characterization of the hydrogels. BSA adsorption and release studies are performed to understand the effect of BSA loading concentration and loading pH on hydrogel performance. Finally, CD-UV spectroscopy analysis is carried out to reveal if BSA maintains its folded structure, which is critical for its biological function, and in vitro wound healing analysis is performed to understand if hydrogel films have a toxic effect on cell migration.

### 3.1. Analysis of Interactions between Components at Molecular Level

#### 3.1.1. Electrostatic Surface Properties of the Dynamic Serum Albumin

Electrostatic surface properties of the serum albumin at different pH values are expected to affect the loading of the protein into the hydrogel as well as its release from it. From full-atom molecular dynamics (MD) simulations, root mean-squared deviation of the structure is calculated (Appendix A). Here, large structural deviation in the first 25 ns indicates an equilibration period for the simulations. This part of the trajectory is thus discarded, and all analyses are carried out based on the last 30 ns. Mean-squared fluctuations of amino acids in physiological conditions (310 K, 150 mM) agree with the experimental B-factors and underline high flexibility of the structure (Appendix A). We select three different snapshots from the MD trajectory, namely at the initial time 0 ns, 22 ns, and 55 ns. For each structure, the electrostatic surface distribution of the structure is calculated using PROPKA and APBS, as detailed in the experimental section. At a pH of 6.4, albumin structure has an unequal charge distribution (Figure 1a). Dynamics of a heart-shaped albumin structure reveal a large positively charged patch between its IB and IIIB domains that opens and closes in solution. The remainder of the solvent accessible surface is mostly negatively charged. At a pH of 4.7, which is the isoelectric point of the albumin structure, the electrostatic surface changes, and a relatively more balanced charge distribution is observed (Figure 1b). Here, sizes of both positive and negative patches are noted to change during the conformational rearrangements. At a lower pH value of 4.3, the entire protein surface is positively charged (Figure 1c). These findings from the theoretical calculations are used to interpret the experimental findings for the protein loading and release experiments.

#### 3.1.2. Zeta Potential Measurements

Zeta potentials (ζ) of pectin (P), pectin–glycerol (P-Gly), and pectin–glycerol–zeolite (PZ-30) solutions are −46 ± 0.3 mV, −49.8 ± 0.7 mV, and −61 ± 0.21 mV, respectively (Figure 2). Since solutions with ζ values greater than +25 mV or less than −25 mV have high degrees of stability, we conclude that all solutions are significantly stable [41]. Negative ζ values are attributed to the high ionization degree of carboxylic moieties (–COOH). The addition of glycerol to pectin solution does not notably alter the zeta potential value of the solution. On the other hand, zeolite inclusion significantly increases the negative ζ, implying the electronegative structure of zeolite particles. On the other hand, the ζ value of M6.4-100 (−41.9 mV) is higher than that of M4.7-100 (−56.6 mV). The lower negative ζ of M4.7-100 is plausibly due to the higher net positive charge of bovine serum albumin (BSA) molecules and less ionic structure of pectin chains, with a pKa of 3.5.

As expected, M4.3-100 (−53.8 mV) has a comparable ζ value to M4.7-100. The electronegativity of M6.4-100 solutions is lower than the M6.4-50 solutions, where a high amount of BSA loading to the film can result in charge neutralization. Results are in accordance with the adsorption studies that will be discussed in Section 3.3.2.

#### 3.1.3. Rheological Analysis

To understand the relationship between structure and properties of the hydrogels, rheological analysis is carried out for two types of samples: pectin suspensions for hydrogel synthesis and cross-linked wet hydrogels. Although no chemical reaction occurs between the components, changes in viscosity and elasticity can be due to electrostatic and hydrogen bond interactions between the system components [25]. In our previous studies, we demonstrated that pH [29] and amount of zeolite [25] can influence the rheological properties of this type of system.

Complex viscosity (η) of the hydrogel suspensions, storage modulus (G’), loss modulus (G’’), and damping factor (tan δ) of the hydrogel films are presented in Figure 3 as functions of radial frequency. η value of all samples decreases linearly with frequency, showing the general shear thinning behavior of the suspensions (Figure 3a). To reveal if pH and BSA concentration have a significant effect on η values of the samples, two-way ANOVA with Tukey’s test is performed at constant shear rates of 1 rad/s and 100 rad/s. At both shear rates, both pH and BSA loading concentration have a significant effect on η values, with *p*-value < 0.05. According to Tukey’s test, which compares sample means, samples prepared at pH 4.3 have significantly higher complex viscosity when compared to samples at pH 4.7 and 6.4 (*p*-value = 0.004 at both rates). With the presence of BSA, complex viscosity decreases significantly (*p*-value = 0.007 at 1 rad/s; *p*-value = 0.005 at 100 rad/s). This result also supports that BSA molecules mostly interact with crosslinking agent Ca^2+^, as will be discussed later.

Figure 3b,c show the frequency sweep tests (mechanical spectra) of hydrogels. Since G’ and G″ are frequency-dependent, and G’ is significantly higher than G″ without any crossover, the hydrogels have highly interconnected gel-like network structures with typical solid-like and mostly elastic behavior. Due to the decomposition of zeolite particles in acidic conditions, G″ of PZ-4.3 [25] has a significantly high value among all hydrogels, and PZ-4.3 solution does not follow the trends; therefore, we do not consider the rheological analysis of this formulation.

PZ-6.4 [25] has higher G’ value than PZ-4.7 coded hydrogels. The network structure is partly loose around pH 4.7. At pH 4.7, pectin–zeolite hydrogel is possibly composed of ordered linear chains, for which aggregation is limited due to the protonation level of carboxylic acid groups in acidic conditions. On the other hand, at pH 6.4, pectin chains form aggregates because of the increasing amount of charges; consequently, a high storage modulus is observed. This was previously discussed in [42] as explaining the effect of pH on protein–protein interactions.

For a more sensitive comparison of the samples, the damping factor, which is an indicator of the presence, position, and relative magnitude of transitions, is calculated for each sample (Figure 3c). To understand if the effects of pH and BSA concentration are important, two-way ANOVA followed by Tukey’s test is performed for the results at two different shear rates, 1 and 100 rad/s. Accordingly, the change in the damping factor with pH change is not significant with a *p*-value, >0.05. The effect of leading BSA concentration is more evident, with a *p*-value < 0.05. Although Gʹ of PZ-6.4 hydrogel is decreased with the addition of BSA (M6.4-100) due to antagonist effect, there is a significant decrease in tan δ when PZ-6.4 and M6.4-100 are compared (*p*-value = 0.001). Lower tan δ value represents a higher stiffness and a stronger intermolecular network. Decrease in tan δ value with BSA addition is an indication of relatively strong physical-chemical interactions for M6.4-100 hydrogel and can be explained by the additional cross-linking behavior of albumin molecules at pH 6.4. It is known that attractive interactions between positively charged peptides/proteins and anionic polysaccharides can lead to gelation [43], where the cationic fragment acts as a crosslink [44].

### 3.2. Structural and Morphological Analysis of Hydrogels

#### 3.2.1. FT-IR Analysis

To confirm the structure of the pectin–zeolite hydrogel and the interactions between pectin, zeolite, and BSA, hydrogel films are analyzed by FT-IR spectroscopy. The spectra obtained for hydrogel films prepared by using different conditions are given in Figure 4. A -COOCH_3_ ester peak belonging to pectin chains is obtained at about 1740 cm^−1^. The other characteristic peaks of pure low-methoxyl pectin comprise the peaks at 1615 cm^−1^ for COO^−^ groups and at 1410 cm^−1^ for C-OH stretching of the carboxylic group [25,45]. For the low-methoxyl pectin–zeolite hydrogel, peaks appear in the 1150–850 cm^−1^ region due to Si-O-Si and Al-O-Al stretching vibrations [46]. For all hydrogels, peaks belonging to pectin and zeolite exist only with minor shifts, indicating that zeolite particles are successfully incorporated into the polymer matrix.

BSA surface exhibits two characteristic bands of protein amide at 1656 (amide I) and 1540 cm^−1^ (amide II), corresponding to the C=O stretch and to the N-H bend coupled with the C-N stretching mode, respectively, which is related to the secondary structures of protein. The existence of BSA in hydrogels in both A6.4-100 and A4.7-100 is confirmed with amide I and amide II bands. The FT-IR spectra of samples prepared by mixing and adsorption methods at pH 6.4 and pH 4.7 (PZ-A6.4-100, PZ-A4.7-100 and PZ-M6.4-100) also show the characteristic peaks of BSA at about 1656 cm^−1^ and 1540 cm^−1^. These results confirm that BSA is loaded to pectin–zeolite hydrogels under both pH 6.4 and 4.7 conditions. On the other hand, the FT-IR spectra of adsorption film prepared at pH 4.3 (A4.3-100) have no amide I and amide II bands. The results are in accordance with the BSA adsorption to pectin–zeolite films that will be described in Section 3.3.2.

#### 3.2.2. AFM Analysis

The surface topology of selected hydrogel samples and their mean surface roughness (SR) are given in Figure 5 and Figure 6, respectively. Pectin hydrogel film inherently has a rough topology. In addition, the preparation of the polymeric film in a solvent medium provides a rough surface due to the exit pathways created by the evaporating solvent [47]. According to our results, increasing the amount of zeolite in hydrogels increases the mean SR. Mean SR of the hydrogel containing 30 mg zeolite (PZ-30) is determined as 49.62 nm. This value increases to 60.21 nm for the hydrogel containing 50 mg zeolite (PZ-50). This result is in accordance with our previous findings from SEM analysis [25], in which zeolite inclusion in pectin films indicated the presence of wider pores when compared to pectin hydrogels without zeolite. Zeolite addition can also increase the crystallinity of pectin hydrogels [25].

The average surface roughness of PZ-30 decreases from 50 nm to 26 nm when BSA is adsorbed at pH 4.7 (M4.7-100). However, it is equal to 78.9 nm for M6.4-100 prepared at pH 6.4. Since pKa of pectin is 3.5, the increasing number of COO^−^ groups with increasing pH leads to the formation of a higher amount of ion bridges with Ca^2+^, resulting in the increase in roughness of the hydrogel surface. A higher degree of roughness provides an extended surface area for protein adsorption [47]. At pH 6.4, protein adsorption is higher than at pH 4.7 and pH 4.3 (Figure 6a).

In Figure 6b, surface roughness is plotted against the concentration of albumin in the hydrogel films prepared with the mixing method, in order to reveal the correlation between roughness and protein adsorption. A high correlation is obtained (R^2^ = 0.9915). For M6.4-x hydrogels, the roughness of the films is proportional to the albumin concentration in the films. As will be discussed in detail in Section 3.3.2, albumin molecules possibly behave as an additional cross-linker at pH 6.4 for PZ films; thus, the porosity of hydrogels may increase with the inclusion of BSA. The surface roughness of the hydrogel films prepared with the adsorption method (A6.4-100) is smaller than that of the sample prepared with the mixing method (M6.4-100). Albumin adsorption occurs after gelation of pectin chains with Ca^2+^ ions. Here, the albumin is expected to bind either to free COO^−^ groups or free Ca^2+^ ions.

With increasing surface roughness, extended surface area can be revealed, which is critical for protein adsorption. Besides providing high surface areas, PZ surfaces accommodate various topological features, such as wide or narrow and deep or shallow valleys and hills. Here, we also investigate the effect of topological distribution on albumin adsorption by analyzing either the height or the position of the hill on hydrogel surfaces using AFM and Gwyddion programs. The height distribution on topographic images can be obtained for a sample surface, showing frequency distribution of pixels at specific heights. The height distributions for PZ-30, PZ-50, M6.4-50, M6.4-100, and M4.7-100 hydrogel films are given in Figure 7. Hydrogel samples have different distributions with various shapes, indicating different topological surfaces. It can be clearly seen from the figure that both M6.4-50 and M6.4-100 samples have bell-shaped distributions, which is evidence for more random distribution of hill heights on the hydrogel film surface. Based on this analysis, we can assume that adsorption zones are homogeneously distributed over the hydrogel surface, which can help to preserve the 3-D structure of albumin molecules. As will be discussed in Section 3.4.3, albumin is released from both M6.4-100 and A6.4-100 hydrogel films by maintaining its folded structure. Here, normal distribution of roughness can be among the reasons for preserving albumin structure.

#### 3.2.3. Scanning Electron Microscopy Analysis

Scanning electron microscopy (SEM) images of the cross-sections of dry films prepared with different methods (mixing and adsorption), pH (6.4 and 4.7), and initial concentration of BSA (25, 50 and 100 mg g^−1^ film) are shown in Figure 8. Figure 8f shows that the LM-pectin film with zeolite is smooth with an irregular porous structure, which can favor solvent accommodation and BSA release. When BSA is added to the pectin films, the films maintain their smoothness, but some small particles are observed (Figure 8a–d). The initial concentrations of BSA and pH significantly affect the morphology of BSA- loaded hydrogel films. An increase in the amount of BSA molecules in the hydrogels leads to a notable change in the film morphology. When a low amount of BSA is used, small BSA particles spread out in the hydrogel (Figure 8d,e). On the contrary, when 100 mg BSA g^−1^ film is used in the films, SEM images of PZ-A6.4-100 and PZ-M6.4-100 hydrogels reveal porous morphology with some holes, as also observed in Figure 8b,c. BSA molecules are observed in SEM images of PZ-A6.4-100 and PZ-M6.4-100 films. PZ-A4.7-100 hydrogel exhibits homogenous and crackly morphology. In general, the differences observed in morphology among pectin–zeolite films with and without BSA and different samples may be an indication of protein diffusion into the channels of pectin hydrogel. We can clearly say that morphological characterization confirms the presence of proteins in the pectin matrix.

#### 3.2.4. Water Contact Angle Measurements

The wettability of dry films with BSA is determined by measuring the contact angles using direct image analysis of sessile drops (Figure 9). While the water contact angle (WCA) of pectin hydrogel film without zeolite is 55° [27], this value shifts to 77° and 100° with the addition of zeolite and BSA to the hydrogel, respectively. Results are in agreement with zeta potential measurements. WCA of the samples prepared with the adsorption method is smaller than that of the samples prepared with the mixing method. For the samples prepared with the adsorption method, due to the smaller quantity of free COO^−^ present after gelation with Ca^2+^, BSA plausibly keeps its total charge. The higher charge on the surface of the hydrogel can interact with water through hydrogen bond formation; this in turn increases the surface wettability and causes a decrease in the contact angle. The small WCA value of PZ-A6.4-100 (88°), when compared to that of PZ-A4.7-100 (99°), can be attributed to the higher content of polar -COO^−^ groups at pH 6.4, which can increase the number of contacts with Ca^2+^ and also increase the porosity of the hydrogel. WCA depends on the surface area and polarity of a material [48]. Similarly, hydrogel at pH 4.7 has a less porous structure. These results agree with the zeta potential calculations, rheology, and swelling (that will be discussed in Section 3.4.1.) experiments.

### 3.3. BSA Loading

#### 3.3.1. BSA Adsorption to Zeolite Particles

First, adsorption of BSA molecules to zeolite particles is examined, where zeolite forms the core part of hydrogel films that are prepared using membrane diffusion method. Experiments are performed using BSA solutions with different concentrations (10, 25, 50, 75, and 100 mg BSA/30 mg zeolite) at various pH values (pH 6.4, 4.7, and 4.3). As shown in Figure 10a, at pH 6.4 and 4.7, approximately 100% of BSA is loaded to zeolite-A for all initial concentrations of BSA. On the other hand, at pH 4.3, which is below the isoelectric point (pI) of BSA (pH 4.6), adsorption efficiency decreases. At the lowest pH value, the adsorption capacity decreases with increasing initial concentration of BSA. Since adsorption is more effective above the pI of BSA, the adsorption is highly affected by electrostatic repulsion. Results are in accordance with the literature [49]. The electrostatic surface calculations (Figure 1) help to understand how a negatively charged BSA at pH 6.4 can be adsorbed onto a negatively charged zeolite surface. The asymmetric charge distribution on the BSA surface and amphoteric character that contains both positive and negative charges plausibly allow their adsorption on zeolite particles, although the total charge of BSA is −7e at pH 6.4 [50].

#### 3.3.2. BSA Adsorption in Pectin–Zeolite Hydrogel Films

The efficiency of BSA adsorption in selected pectin–zeolite films is given in Table 1 and Figure 10b. Both at pH 6.4 and 4.7, BSA molecules are successfully adsorbed in pectin–zeolite matrices, as implied by FT-IR, SEM, zeta potential, and rheology analyses. BSA uptake at pH 4.3 is low. Two-way ANOVA and Tukey’s test to assess the effect of pH and BSA concentration on BSA uptake are conducted. We show that BSA adsorption efficiency of pectin–zeolite hydrogel films significantly decreases with decreasing pH of the loading medium (*p* < 0.0001). The effect of BSA loading concentration is also found to be significant, with a *p*-value = 0.027. A6.4-100 and A6.4-50 adsorb 88 mg BSA/g film and 37 mg BSA/g film, respectively, whereas A4.7-50 film can adsorb only 20 mg BSA/g film. Although P-A4.3-50 film prepared without zeolite particles can adsorb 21 mg BSA/g film, surprisingly, BSA cannot be adsorbed by A4.3-50, most probably due to the structural decomposition of zeolite particles under acidic conditions. We also examine the swelling ratios of the hydrogels during adsorption (Figure 10). At pH 6.4, an increasing amount of adsorbed BSA in the hydrogels causes a decrease in the swelling rate of the hydrogels. At this pH, the highest swelling ratio is obtained for the BSA-free buffer solution. This can be explained by the additional crosslinking behavior of BSA, resulting in reinforcement and permeability decrease in the pectin–zeolite hydrogel [43]. However, hydrogel in BSA-free solution swells less than the sample in buffer solution with 100 mg BSA, since at pH 4.7, a lower amount of COO^−^ groups in pectin chains would be available in an acidic environment. This difference can be attributed to the lower amount of COO^−^ groups available in an acidic environment in pectin chains resulting in less pronounced attractive electrostatic interaction of COO^−^ groups of pectin and amino groups of the protein.

Based on our findings, loading pH alters (i) the electrostatic distribution of amphoteric BSA surface; (ii) protonation and deprotonation of -COOH groups of pectin [51]; (iii) the extent of interactions between Ca^2+^ and BSA or pectin hydrogel [52]. The interactions between proteins and solid surfaces can be affected by the heterogeneity of the surface and molecular properties of the adsorbed molecules [53]. It is known that the hydrogel films produced by ionic gelation are porous [54]. Pectin gelations at pH 4.7 and pH 4.3 exhibit a less intertwined and porous network compared to those at higher pH, such as pH 6.4, as shown by the SEM analysis (Figure 8).

### 3.4. BSA Release

#### 3.4.1. Swelling Behavior of BSA-Loaded Hydrogels

The equilibrium swelling ratios of selected hydrogel films with BSA are shown in Table 1 and Figure 11c. Swelling behavior of the hydrogels are significantly affected by pH, concentration of BSA, and preparing methods, as assessed using the ANOVA method with Tukey’s test with *p*-value < 0.0001 for each parameter. The films prepared with adsorption (A6.4-50) method display much higher swelling ratios (49.2) when compared to the films prepared with the mixing method (M6.4-50; 29.9) (*p*-value < 0.0001). In the mixing method, all albumin is loaded into pectin–zeolite mixture before gelation, whereas in the adsorption method, dry pectin–zeolite hydrogel films (A6.4-50) can only adsorb 74% of the initial albumin concentration. Tukey’s pairwise comparison indicates a *p*-value of 0.00001. Albumin has reinforcement and additional crosslinking effects on pectin–zeolite hydrogel films, as was described above. Thus, the swelling degree of mixing films decreases with increasing amount of BSA molecules loaded into the hydrogel. When more BSA binds to film, the film becomes more compact, leading to a lower degree of swelling. The higher swelling degree of A6.4-50 hydrogel film, when compared to that of A4.7-50, is possibly due to a lower amount of BSA adsorption and less ionic character of pectin chains at pH 4.7. These findings agree with contact angle measurements (Figure 9).

We also examine the swelling ratios of the hydrogels during adsorption (Figure 11c). For this purpose, we prepare pectin–zeolite (PZ-30) hydrogels and put them into buffer solutions containing different amounts of BSA (25, 50 or 100 mg/g film) at two different pH (6.4 or 4.7) values. At pH 6.4, an increasing amount of adsorbed BSA in the hydrogels causes a decrease in the swelling rate of the hydrogels. At this pH, the highest swelling ratio is obtained for the BSA-free buffer solution. This can be explained by the additional crosslinking behavior of BSA, resulting in the reinforcement and decrease in permeability of pectin–zeolite hydrogel [43,44]. However, hydrogel in BSA-free solution swells less than the sample in buffer solution with 100 mg BSA, since a lower amount of -COO^−^ groups is available in pectin chains in the acidic environment with pH 4.7. This difference can be due to a lower amount of -COO^−^ groups available in an acidic environment in pectin chains and a lower amount of attractive electrostatic interactions between -COO^−^ groups of pectin and amino acids of the protein.

#### 3.4.2. BSA Release from Zeolite Particles and Hydrogel Films

BSA is irreversibly adsorbed on zeolite particles (Figure 10a and Figure 11a). This irreversible adsorption behavior is plausibly related to the hydrogen bond interactions between zeolite surface and albumin. Here, water molecules may be also involved at the protein-surface interface, where they can simultaneously interact with protein and zeolite surfaces.

Based on the amount of BSA released from the selected pectin–zeolite and pectin hydrogels in pH 6.4 TRIS buffer, the released amount is significantly affected by the loading method (adsorption or mixing, with *p*-value < 0.0001) and initial concentrations of BSA (*p*–value < 0.0001) (Figure 11b). We observe that zeolite addition to pectin hydrogels significantly increases BSA releases both in adsorption (P-A6.4-50, P-A6.4-100, P-A4.3-50) and mixing (P-M6.4-75 and P-M6.4-100) samples (with *p*-value = 0.008) (Figure 11b). At low initial concentrations of BSA (A6.4-10; 10 mg/g film, A6.4-25; 25 mg/g-film), although approximately all BSA is adsorbed in the pectin–zeolite films, no release is observed. This can be attributed to an insufficient driving force for the diffusion of BSA molecules out of the films, i.e., lower initial protein concentration. On the contrary, A6.4-50 and A6.4-100 hydrogels release most of the adsorbed protein to the dissolution media. ANOVA method with Tukey’s test also supports this observation; release from the low-BSA-concentration (10 and 25 mg/g film) samples are significantly different from the samples with high concentration (50, 75, 100 mg BSA/g film) with *p*-value < 0.0001. Hydrogel films prepared with mixing method exhibit significantly low release ratio when compared to the adsorption films (*p* < 0.0001). M6.4-100 and M6.4-75 films achieve releasing 41 mg/g film and 23 mg/g film BSA, respectively, which correspond to about 40% and 30% release of immobilized BSA. On the other hand, almost all adsorbed BSA can diffuse from the films prepared with the adsorption method. The different release behaviors of the samples prepared with the adsorption and mixing methods seem to be related to the adsorbing hydrogel surface. When considering adsorption of BSA in pectin–zeolite films, during the diffusion throughout the hydrogel, BSA molecules encounter pectin chains first. However, in the case of mixing film, BSA is primarily loaded on the zeolite, and then BSA-loaded zeolite acts as a core membrane mixed with pectin solution.

#### 3.4.3. Circular Dichroism (CD) Measurement Analysis

To understand to what extent adsorbed albumin can preserve its structural features after being released from the hydrogel samples to the external solution, circular dichroism (CD)-UV spectroscopy is used. CD can determine the putative secondary structure alterations of the released BSA in the external buffer solution [35]. We monitor the effects of pH (8.2 and 6.4) on the structural change of BSA in hydrogels, which are prepared with either mixing or adsorption methods (Figure 12). BSA has two negative bands in the far-UV region at around 208 and 222 nm, which represent the alpha-helix structure of the protein and the aromatic amino acid residues, respectively [35,55], with strong ellipticity values confirming the presence of characteristic features of mostly alpha-helical-type protein secondary structures [35]. Figure 12 indicates that BSA molecules cannot preserve their secondary structures while being released from the M8.2-100 and M6.4-75 hydrogel films. On the other hand, at pH 6.4, 100 mg BSA-containing films (A6.4-100, M6.4-100) gave the characteristic alpha-helical peaks for BSA, showing that the secondary structures are maintained. According to literature, the interaction between BSA and metal complexes often leads to a perturbation of the secondary structure of the protein [51], leading to a partial loss of alpha-helix conformation. Ca^2+^ ions can bind BSA at the pH range 6.5–8.5 for both mixing and adsorption films that have 100 mg BSA (A6.4-100, M6.4-100). Films loaded at the pH value of 6.4 preserve the secondary structure of proteins while the protein is released to the external buffer solutions. On the contrary, the films prepared at a pH value of 8.2 (M8.2-100) do not give the characteristic alpha-helix bands of BSA, which can be attributed to the higher mobility of pectin chains due to the higher ion–ion repulsion and the possible presence of Ca^2+^-BSA interactions at this relatively high pH value.

Hydrogel films with relatively low amounts of BSA (75, 50, and 25 mg BSA/g film) prepared at pH 6.4 yield chiral BSA peaks. This different binding character can be explained by the more mobile environment provided by the lower amount of BSA molecules. The chiral CD pattern of these films needs to be detailed in a further study.

### 3.5. In Vitro Wound Healing Assay

The wound healing properties of the hydrogels are evaluated by in vitro wound healing assay. It is shown that none of the hydrogels inhibited cell growth and migration of fibroblast cells during the 48 h and 72 h incubation periods (Figure 13). When we measure the gap area, we observe that the cells that are treated with hydrogels tend to migrate to wound site, similar to the control group.

## 4. Conclusions

In this study, we successfully designed novel hydrogels based on low-methoxy pectin and NaA zeolite particles as a wound dressing material for either albumin adsorption from wounds with excess exudate or controlled albumin delivery to the wounded area for patients with hypoalbuminemia. While the low cytotoxicity of the pectin–zeolite-based hydrogels was previously demonstrated [25], here, we particularly focused on preserving albumin’s conformation while it is released to external buffer solutions to maintain its biological function.

Here, zeolite particles behave as core membranes. Protein loading efficiency to zeolite particles decreased at pH 4.3. Both FT-IR and albumin loading experiments revealed that, at pH 6.4 and 4.7, albumin molecules were successfully loaded to pectin–zeolite hydrogel films. On the other hand, at pH 4.3, hydrogel films could not adsorb albumin molecules. This can be attributed to the decomposing structure of zeolite particles in an acidic medium.

Roughness is a crucial parameter in protein adsorption, as it supplies more adsorption sites on the hydrogel surface. We evaluate the mean surface roughness of the hydrogels together with their topological surfaces, such as the height distribution of hills. Results showed that at pH 6.4, surface roughness increased with albumin inclusion in the pectin–zeolite hydrogel. SEM images pointed out the differences in morphology of pectin–zeolite hydrogel with and without albumin, as well.

Looking at both the swelling ratio and drug release results of the hydrogels, we suggest that albumin release was dominated by swelling. Rheological analyses revealed that at pH 6.4, the damping factor decreased with the addition of albumin molecules, indicating a strong intermolecular network, due to the interaction between the positively charged regions on the protein surface and the anionic groups of pectin chains. In this way, albumin molecules possibly behaved as additional cross-linker agents for pectin–zeolite hydrogel films. Circular dichroism analyses indicated that films prepared with either adsorption or mixing methods (M-6.4-100 and A6.4-100) could preserve albumin’s folded structure while releasing, indicating that hydrogels synthesized under these conditions can preserve the biological activity of albumin.

## Figures and Tables

**Figure 1 polymers-14-00460-f001:**
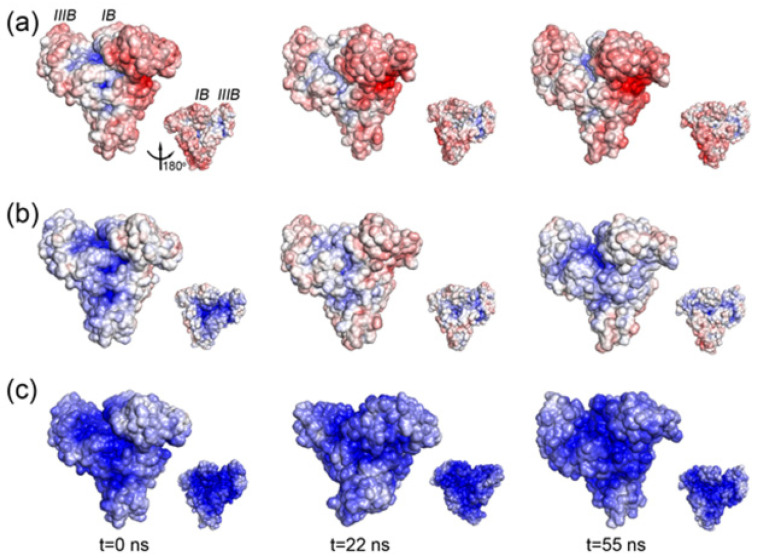
Surface charge distribution for different conformers of albumin (**a**) at pH 6.4; (**b**) at pH 4.7; (**c**) at pH 4.3. Each protein surface is colored as red, white, blue, corresponding to −10 kT/e, 0 kT/e, and +10 kT/e, respectively. Small-sized cartoon representations display a different perspective, as indicated for one structure at t = 0 ns.

**Figure 2 polymers-14-00460-f002:**
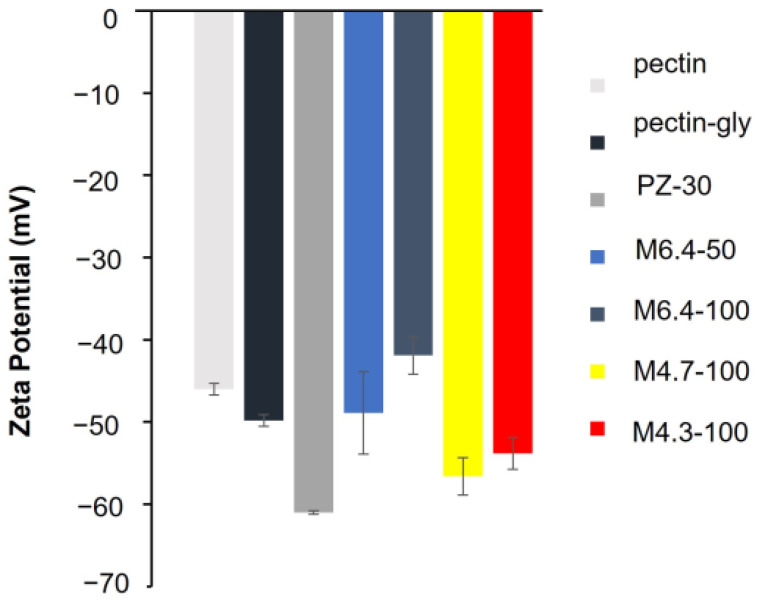
Zeta potential values of hydrogel samples in solution.

**Figure 3 polymers-14-00460-f003:**
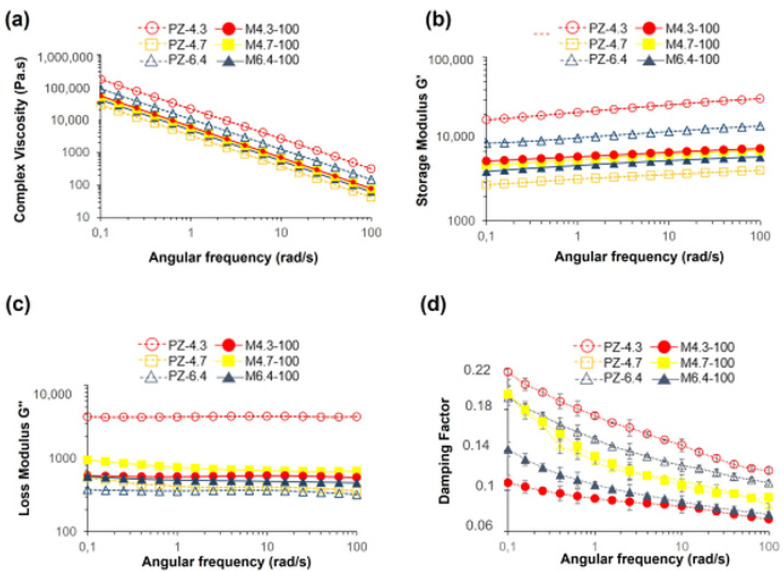
Rheological analysis of the hydrogels; (**a**) complex viscosity, (**b**) storage modulus G’, (**c**) loss modulus G’’, and (**d**) damping factor.

**Figure 4 polymers-14-00460-f004:**
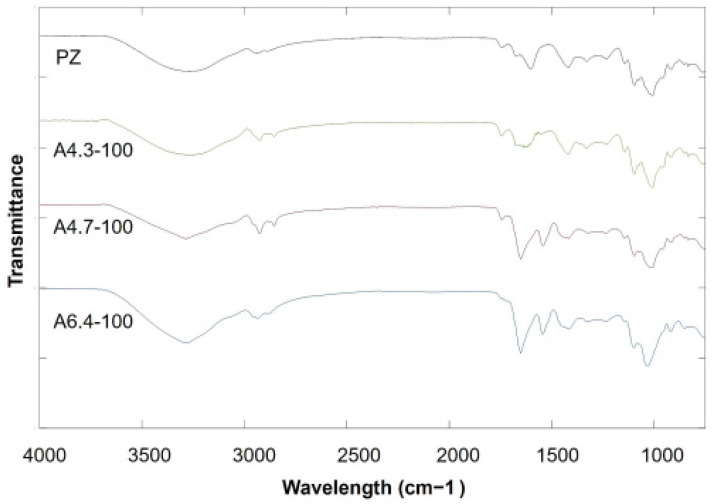
FT-IR spectra of the hydrogels.

**Figure 5 polymers-14-00460-f005:**
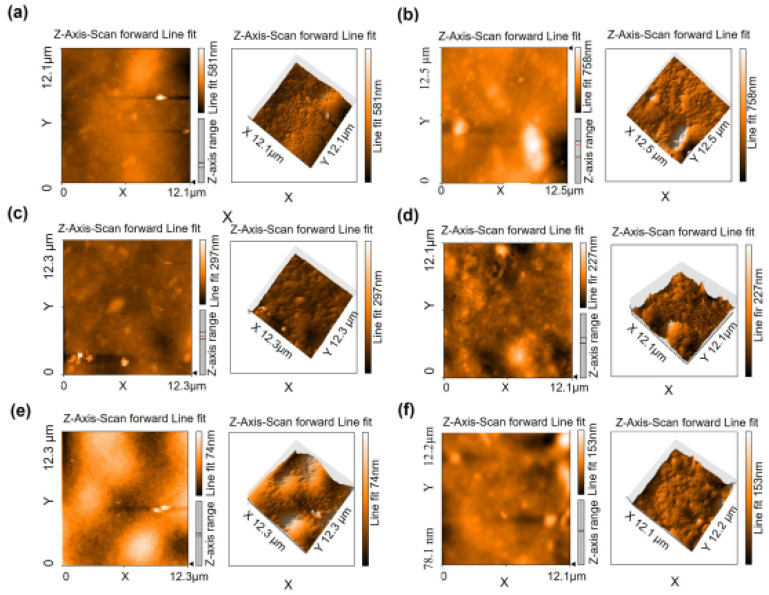
Surface topography images of dry hydrogel samples; (**a**) PZ-30, (**b**) PZ-50, (**c**) M6.4-50, (**d**) M6.4-100, (**e**) A6.4-100, (**f**) M4.7-100.

**Figure 6 polymers-14-00460-f006:**
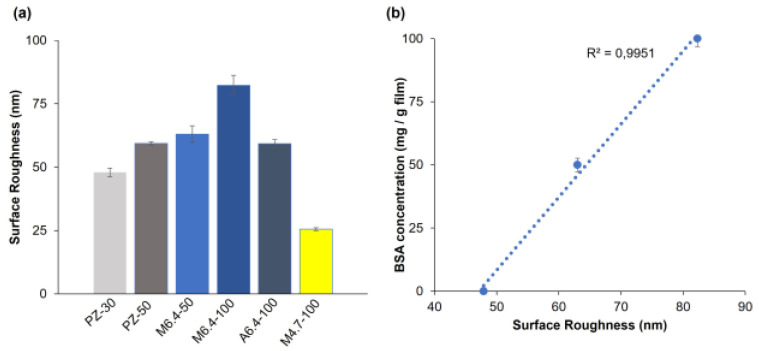
(**a**) Surface roughness of the hydrogel samples, (**b**) variation of surface roughness of M6.4-x hydrogels with BSA concentration.

**Figure 7 polymers-14-00460-f007:**
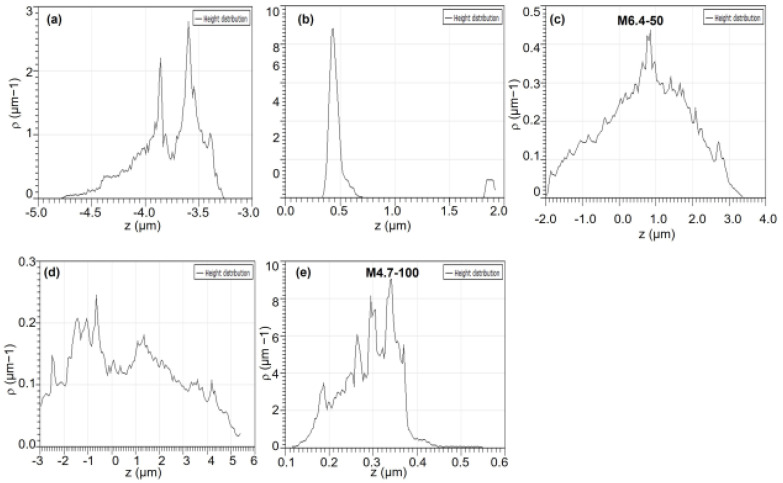
Topological distributions of dry hydrogel samples: (**a**) PZ-30, (**b**) PZ-50, (**c**) M6.4-50, (**d**) M6.4-100, and (**e**) M4.7-100.

**Figure 8 polymers-14-00460-f008:**
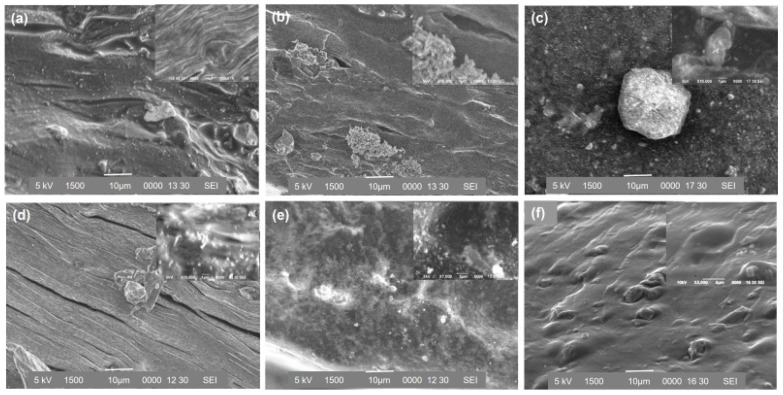
SEM images of BSA-loaded hydrogels: (**a**) A4.7-100, (**b**) M6.4-100, (**c**) A6.4-100, (**d**) M6.4-50, (**e**) M6.4-25, and (**f**) PZ-6.4.

**Figure 9 polymers-14-00460-f009:**
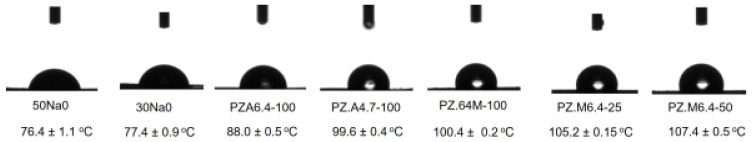
Contact angle images of BSA-loaded hydrogels.

**Figure 10 polymers-14-00460-f010:**
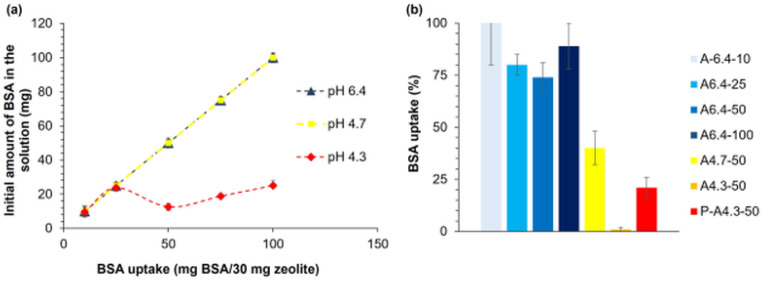
(**a**) BSA uptake as a function of BSA concentration in zeolite at various pH values, (**b**) BSA adsorption efficiency of hydrogel films.

**Figure 11 polymers-14-00460-f011:**
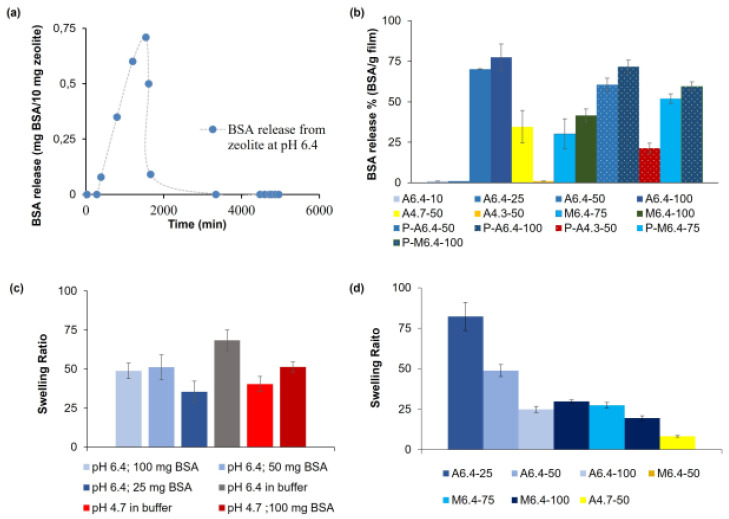
(**a**) BSA release from zeolite particles, (**b**) BSA release from pectin–zeolite and pectin hydrogels, (**c**) Swelling ratios of PZ-30 hydrogel films in different buffer solutions containing BSA molecules, (**d**) Equilibrium swelling ratios of hydrogels.

**Figure 12 polymers-14-00460-f012:**
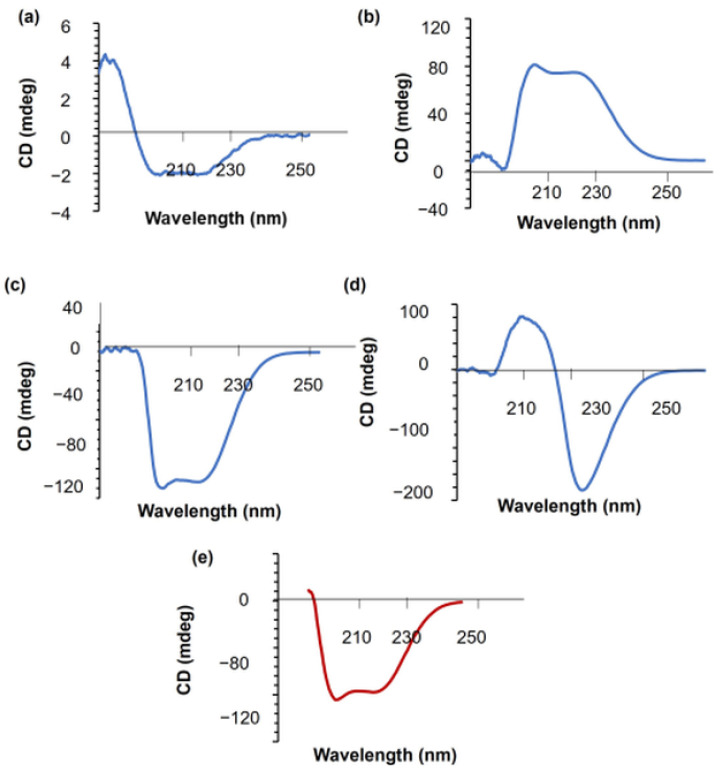
CD-UV analysis of the hydrogels: (**a**) M6.4-100, (**b**) M6.4-75, (**c**) A6.4-100, (**d**) M8.2-100, and (**e**) pure BSA.

**Figure 13 polymers-14-00460-f013:**
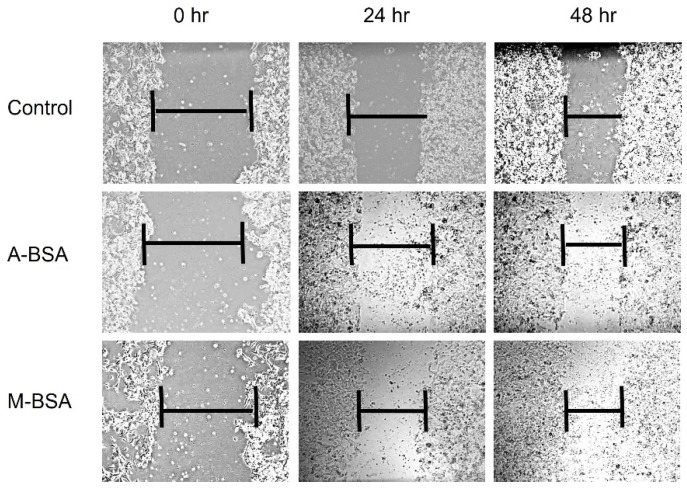
In vitro wound healing analysis of A6.4-100 and M6.4-100 hydrogel films.

**Table 1 polymers-14-00460-t001:** Synthesis conditions, contact angle, swelling degree, and amounts of adsorbed and released BSA from the pectin-based hydrogels.

Code	Method	Zeolite-Na Amount (mg/g film)	Initial BSA Concentration (mg BSA/g film)	pH of BSA Solution	Contact Angle (°)	Swelling Degree	Amount of Adsorbed BSA (mg/g film)	Amount of BSA Release (mg BSA/g film)
PZ-50	-	50	0	-	77	-	-	-
PZ-30	-	30	0	-	76	-	-	-
A6.4-10	Adsorption	30	10	6.4	-	-	10	0
A6.4-25	Adsorption	30	25	6.4	-	79.2	16	0
A6.4-50	Adsorption	30	50	6.4	-	49.2	37	35
A6.4-100	Adsorption	30	100	6.4	88	24.9	88	77
M6.4-25	Mixing	30	10	6.4		-	10	0
M6.4-25	Mixing	30	25	6.4	105	-	25	0
M6.4-50	Mixing	30	50	6.4	107	29.9	50	5
M6.4-75	Mixing	30	75	6.4	-	27.3	75	23
M6.4-100	Mixing	30	100	6.4	100	19.4	100	41
A4.7-50	Adsorption	30	50	4.7	-	8.2	20	17
A4.7-100	Adsorption	30	100	4.7	99	-	50	-
A4.3-50	Adsorption	30	50	4.3	-	-	0	0
P-A4.3-50	Adsorption	-	50	4.3	-	-	21	5

## Data Availability

All data are available upon request.

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
