# Peer review of "Pectin–Zeolite-Based Wound Dressings with Controlled Albumin Release"

_polymers, 2022, doi:10.3390/polym14030460_

Round 1

Reviewer 1 Report

Manuscript number: polymers-1530716-peer-reniew-v1

Article Type: Article

Title: Pectin-Zeolite-Based Wound Dressings with Controlled Albumin Release

The Authors were designed novel hydrogels based on low methoxy pectin and NaA zeolite particles. It would be wound dressing material for either albumin adsorption from wounds with excess exudate or controlled albumin delivery to the wounded area for patients with hypoalbuminemia.

Comments:

Chapter Materials and Methods: in the manuscript no statistical methods, please use it and show for example, statistical differences between results;

Fig. 3: Modulus G’ and G” should have units; Fig. 3b and 3c should have described x axis and units;

Fig 3a: please remove 0 and put 1, 10 and 100 on the x axis, should be the same like for fig. b, c, d

Line 222: … Discussio; should be discussion;

Line 612: Are You sure that the word “hypoalbeneumia” is correct? Is it the same as a “hypoalbuminemia” or “hypoalbuminaemia”

Recommendation: major revision

Author Response

Please see the attchment.

Reviewer 2 Report

The paper Pectin-Zeolite-Based Wound Dressings with Controlled Albu-min Release prepared by Banu Kocaaga, present new and interesting results that deserve to be published after some minor improvements:

  1. figure 3 - please increase the resolution and font size.
  2. figure 4 - please replace yellow with another color, and there are 2 blue spectra, please choose another color for one of them.
  3. figure 5 - please increase the resolution of figure 5, also the text is blurred and is not readable.
  4. figure 6 - please add statistical results
  5. figure 7 - font-size too low, images are blurred.
  6. figure 8 - scale bar is hard to be identified. Please replace it with a new one
  7.  row 495 - there are several bullets. Please remove them and create a story text. 
  8. figure 13 - the figure was cropped by cutting text, check M from M-BSA

Reviewer 3 Report

The manuscript entitled “Pectin-Zeolite-Based Wound Dressings with Controlled Albumin Release.” by Banu Kocaaga shows the preparation of the synthesis of pectin-based hydrogel films, with the ability to deliver albumin as a wound dressing. Well written manuscript and carries considerable merits. While the work is of interest, after addressing the following comments, this manuscript is suitable for publication.

  1. Having a better introduction or comparing the results with other literature is suggested. For wound healing, tissue-engineered construct and wound dressing affects the healing process via multiple factors, e.g., angiogenesis, antibacterial properties, .... Better introduction on skin wound healing and the employed materials suggested; recommended literature:

https://doi.org/10.1002/btm2.10254

https://doi.org/10.1016/j.reth.2021.02.007

  1. The quality of figures 3, and 13 is very low. However, it may be due to the PDF conversion. Please check and use 600 DPI resolution. Also, font size should be in harmony in all figures.
  2. Having better analysis for CD, the following literature may be useful:

https://doi.org/10.2174/092986609788490212

  1. For in vitro wound healing assay I suggested more precise analysis and report the wound area data also.
  2. To determine the cytotoxicity and cell viability I recommended MTT assay or AlamarBlue assay.

Round 2

Reviewer 1 Report

Manuscript number: polymers-1530716-peer-reniew-v2

Article Type: Article

Title: Pectin-Zeolite-Based Wound Dressings with Controlled Albumin Release

The Authors were designed novel hydrogels based on low methoxy pectin and NaA zeolite particles. It would be wound dressing material for either albumin adsorption from wounds with excess exudate or controlled albumin delivery to the wounded area for patients with hypoalbuminemia.

Comments:

Chapter Materials and Methods: in the manuscript no statistical methods, please use it and show for example, statistical differences between results.

To analyze the results more comprehensively, statistical methods are applied for the rheological data and error bars are also placed on Figure 3 and Figure 6.

Reviewer:

“Statistical methods:  In order to find relationship between physochemical and the rheological parameters of the formulations ANOVA method was used”. Could You show me where in the article the ANOVA was used? If is ANOVA, what kind of test did You use?

Recommendation: major revision

Reviewer 3 Report

Remove dash for albumin in the title "Albu-min".

Input only the final figures in the manuscript. Also, please use more readable fonts such as Helvetica or Arial. Please modify the figures throughout the manuscript. Also, some fonts may need to be enlarged to be more readable. Also, font size should be in harmony in all figures. some of them are too big or small. 

Please recheck your references.

Round 3

Reviewer 1 Report

I accept manuscript in present form

Kind regards

Reviewer